# Structural insights into the Venus flytrap mechanosensitive ion channel Flycatcher1

Sebastian Jojoa-Cruz [1,6], Kei Saotome [1,2,4,6], Che Chun Alex Tsui [1,3], Wen-Hsin Lee [1], Mark S. P. Sansom [3], Swetha E. Murthy [2,5✉], Ardem Patapoutian [2✉] & Andrew B. Ward [1✉]

Flycatcher1 (FLYC1), a MscS homolog, has recently been identified as a candidate mechanosensitive (MS) ion channel involved in Venus flytrap prey recognition. FLYC1 is a larger protein and its sequence diverges from previously studied MscS homologs, suggesting it has unique structural features that contribute to its function. Here, we characterize FLYC1 by cryo-electron microscopy, molecular dynamics simulations, and electrophysiology. Akin to bacterial MscS and plant MSL1 channels, we find that FLYC1 central core includes side portals in the cytoplasmic cage that regulate ion preference and conduction, by identifying critical residues that modulate channel conductance. Topologically unique cytoplasmic flanking regions can adopt 'up' or 'down' conformations, making the channel asymmetric. Disruption of an up conformation-specific interaction severely delays channel deactivation by 40-fold likely due to stabilization of the channel open state. Our results illustrate novel structural features and likely conformational transitions that regulate mechano-gating of FLYC1.

[1] Department of Integrative Structural and Computational Biology, Scripps Research, La Jolla, CA 92037, USA. [2] Howard Hughes Medical Institute, Department of Neuroscience, Dorris Neuroscience Center, Scripps Research, La Jolla, CA 92037, USA. [3] Department of Biochemistry, University of Oxford, South Parks Road, Oxford OX1 3QU, UK. [4] Present address: Regeneron Pharmaceuticals, Tarrytown, NY 10591, USA. [5] Present address: Vollum Institute, Oregon Health and Science University, Portland, OR 97239, USA. [6] These authors contributed equally: Sebastian Jojoa-Cruz, Kei Saotome. ✉email: murthysw@ohsu.edu; ardem@scripps.edu; andrew@scripps.edu

Sensing and transduction of mechanical stimuli are essential across all kingdoms of life, and are in large part carried out by mechanosensitive (MS) ion channels that open in response to force[1,2]. A mechanosensory function that has been studied for decades in plants is the rapid closure of the bilobed trap of the carnivorous plant *Dionaea muscipula* (Venus flytrap) in response to touch[3,4]. Deflection of sensory trigger hairs in the trap results in an action potential[4-6], and two action potentials within a short period of time are sufficient for trap closure[7]. Recently, two independent studies have proposed that the gene *Flycatcher1* (*FLYC1*) is one of the candidates involved in Venus flytrap touch sensation, based on 85-fold enrichment of the transcript in sensory trigger hairs relative to trap tissue[8,9]. FLYC1 selectively localizes to mechanosensory cells in trigger hairs, and overexpression of FLYC1 in mechanically insensitive HEK cells results in robust stretch-activated chloride permeable currents, suggesting that FLYC1 encodes a MS ion channel[8]. Furthermore, two FLYC1 homologs have highly enriched transcripts in the sensory tentacle of a different carnivorous plant within the Droseraceae family, *Drosera capensis* (Cape sundew)[8]. Therefore, FLYC1 channel homologs may have a general role in touch-induced prey recognition in carnivorous plants.

FLYC1 is an ortholog of prokaryotic MS ion channel MscS (mechanosensitive ion channel of small conductance), which functions as an osmotic release valve[10,11], and is perhaps the most extensively studied MS channel[12-18]. MscS is part of a large superfamily that includes, among many orthologs across different kingdoms, five orthologs in *E. coli* alone[19], and the eukaryotic plant homologs, MSL (MscS-like) ion channels[20,21]. There is growing evidence that MSL proteins function as MS ion channels in plants. First, MS ion channel activity has been demonstrated for heterologously expressed *Arabidopsis thaliana* (At) MSL1[22], MSL8[23], and MSL10[24]. Second, MSL8 has been shown to be a mechanical stress sensor for pollen hydration and germination[23]. And finally, MSL9 and MSL10 are required for MS ion channel activity in root cells[21]. Recently, the structure of the first MscS eukaryotic homolog, AtMSL1, was solved by cryo-electron microscopy (cryo-EM), revealing structural homology to *E. coli* MscS (EcMscS) that extends beyond the previously termed MscS domain[20,25-29]. MSL1 localizes to the inner mitochondrial membrane and, at a sequence level, it is closer to bacterial homologs than to other plant paralogs that localize to the plasma membrane[20]. The sequence dissimilarity and different intracellular localization serve as evidence of functional diversity and potential structural differences within the MSL family[20]. FLYC1 is also a homolog of *Arabidopsis* MSL proteins and has highest similarity to MSL10 (48% sequence identity)[8], which localizes to the plasma membrane[21]. Furthermore, relative to EcMscS and AtMSLs, FLYC1 has distinct biophysical characteristics including higher chloride selectivity, lower single channel conductance, and lower mechanical threshold, suggesting that these channel properties might be governed by unique structural differences between the three ion channel families[8,21,22]. For these reasons, and its proposed involvement in Venus flytrap mechanosensation, we sought to characterize the structural properties of FLYC1.

Here, we report high resolution cryo-EM studies of Venus flytrap FLYC1, extended by molecular dynamics (MD) simulations and structure-function analysis. Our results provide insights into the structural components responsible for MS channel gating and ion conduction, illuminating the structural and mechanistic diversity within the MscS/MSL protein family across kingdoms of life.

## Results

### Overall architecture of FLYC1 and comparison to MscS/MSL1 homologs.
We expressed a C-terminal EGFP fusion construct of full-length FLYC1 protein in mammalian cells, purified the protein in detergent, and prepared grids for cryo-EM analysis. A subset of particles identified by 3D classification yielded a 2.8 Å resolution reconstruction without symmetry applied. In this density map, the central core of the heptameric particle retained C7 symmetry while the peripheral portions exhibited asymmetry and weaker density, suggesting conformational flexibility. C7 symmetry expansion followed by focused classification and refinement of the peripheral region yielded two distinct classes with improved density. The two classes are referred to as either 'up' or 'down' based on the orientation of two cytoplasmic helices that extend from the transmembrane domain (TMD) (Fig. 1a, b). In our dataset, protomers favored the up over the down class, with approximately 72 and 16% of the symmetry expanded particles assigned to each class, respectively (Supplementary Fig. 1). A composite map was obtained from merging the C1 reconstruction and focused classification maps in a 6 up to 1 down protomer ratio, based on visible density from the C1 reconstruction (Fig. 1a). The molecular model of FLYC1 was initially built from the best-resolved of the two classes, the up class, which was the primary subject of our structural analysis unless otherwise stated (Supplementary Table 1). The model encompasses the majority of the second half of the full-length sequence. However, a significant portion of the protein, including the N-terminus and flexible loops connecting TM helices, were not modeled, or modeled as poly-alanine due to ambiguous density (Supplementary Figs. 2, 3). Interestingly, we observed density resembling a pocket lipid near TM6b, at a location similar to those highlighted in previous reports of EcMscS and AtMSL1 (Fig. 1a)[18,27,30,31]. We conservatively modelled the pocket lipid as palmitic acid to represent one acyl chain of a phospholipid.

FLYC1 comprises a homoheptamer with six TMs (TM1–TM6) per protomer and intracellular N and C termini (Fig. 1a–c). TM1–TM6 are arranged in an approximate single file, with TM6 closest to the central heptad axis (Fig. 1c). A glycine-containing hinge (G575) splits TM6 into two halves (TM6a and TM6b), and the β-rich C-termini of the FLYC1 protomers come together to form a hollow cytoplasmic cage (Fig. 1d, e). The hinged C-terminal TM helix and cytoplasmic cage are also present in structures of MscS/MSL homologs (Fig. 1d, e)[12,16,27], whereas the remainder of the subunit structure has distinct features. These include three additional outer TM helices (TMs 1–3) and a long, partially structured linker between TM4 and TM5 that protrudes into the cytoplasm, resembling an oar. This linker corresponds to the two helices extending from the TMD observed in the density map. Superposition of the C-terminal regions of FLYC1 to previously published MscS/MSL orthologs in various functional conformations[12,15,27,30,32] reveals that TM4–TM6a have an arrangement distinct from all other reported structures. Viewed from extracellular side, the outer TMs of FLYC1 are rotated clockwise relative to those in other structures (Supplementary Fig. 4a). Notably, the closed forms of AtMSL1 and EcMscS are discernibly more similar to each other than FLYC1 is to either (Supplementary Fig. 4a). Qualitatively, the TM4–TM6 conformation of our current FLYC1 structure appears to be most similar to the open form of EcMscS[15]. As such, the cytoplasmic ends of TM4–TM5 (EcMscS TM1–TM2) interact with TM6b (TM3b) of the same subunit[30], instead of the adjoining subunit as occurs in the closed state of EcMscS and AtMSL1 (Fig. 1e and Supplementary Fig. 4a). This conformation appears to be stabilized by the presence of bulkier side chains, instead of glycine and alanine, at the interface between TM6a of different subunits, resulting in a looser and more upright packing of the transmembrane helices relative to EcMscS and AtMSL1 (Supplementary Fig. 4b). The looser packing causes a widening of the pore towards the extracellular side and a larger surface of polar residues to be exposed and lining the pore in the TMD relative to

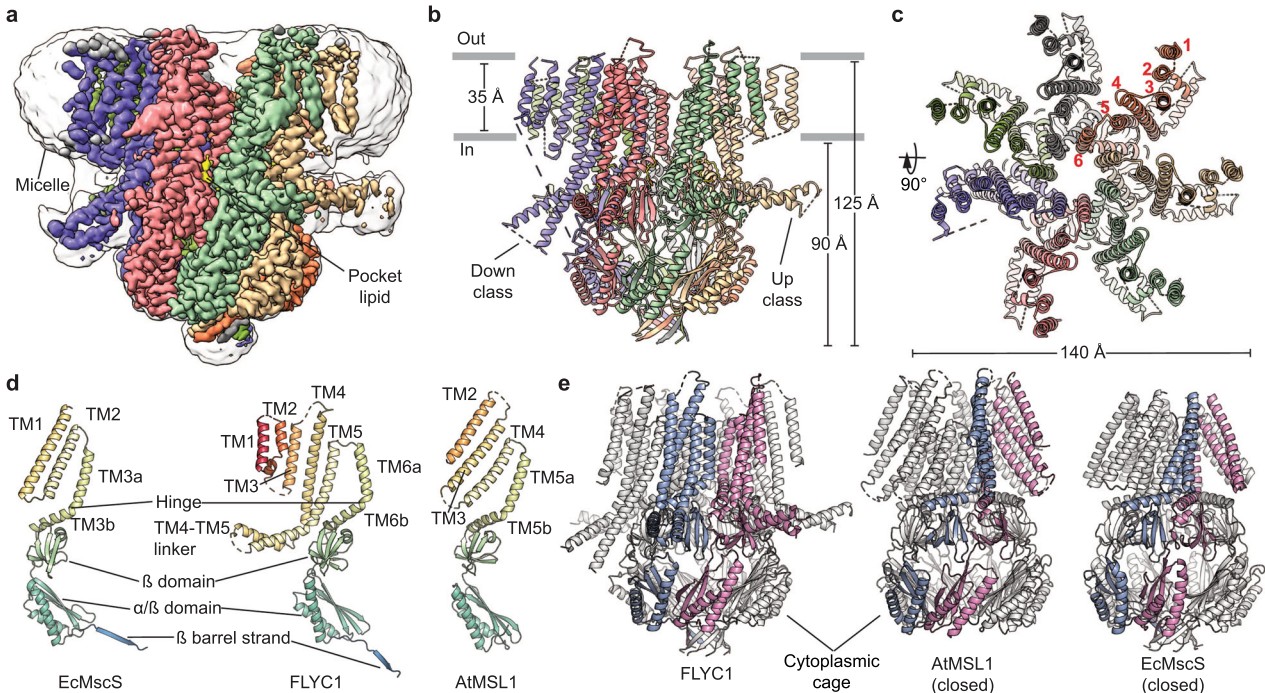

**Fig. 1 FLYC1 structure. a** Cryo-EM density of FLYC1 colored by subunit and enclosed in detergent micelle (unsharpened map gaussian-filtered to 1.5 σ). Side (**b**) and top (**c**) view of FLYC1 model. In top view, TM helices are labeled from 1 to 6, according to their position in the amino acid sequence. **d** Cartoon representation of protomers of EcMscS (PDB: 2OAU, closed state), FLYC1, and AtMSL1 (PDB: 6VXM, closed state). **e** Side views of FLYC1 heptamer and its homologs, with two protomers colored blue and pink, respectively. Rotation of TM helices of each protomer relative to conserved cytoplasmic cage is limited for FLYC1 relative to AtMSL1 and EcMscS, where TM helices interact with cytoplasmic domain of the contiguous subunit. TM1–TM2 of FLYC1 are not shown for simplicity.

EcMscS (Fig. 2a, b and Supplementary Fig. 4c). Despite its apparent structural similarity at the protomer level to the open state of EcMscS, we do not believe our current structure represents a fully open form of FLYC1, as outlined below.

**A phenylalanine ring potentially forms the pore gate of FLYC1.** The central pore axis of FLYC1 is lined by TM6 in the TMD and by the cytoplasmic cage in the cytosol. The narrowest constriction in the TMD is created by a ring of phenylalanine side chains (F572) at the bottom of TM6a (Fig. 2a). The pore has a minimum van der Waals radius of ~3.5 Å in this region, larger than the radius of a hydrated chloride ion (2.5 Å)[33] (Fig. 2b). Though this may suggest the pore can conduct ions in this conformation, the hydrophobicity of the F572 ring could in principle present an energetic barrier to ion flow[34]. Indeed, the original EcMscS structure[12], where residues L105 and L109 form a hydrophobic gate of similar width in this region (Fig. 2b), was initially interpreted as open but has since been described as closed/inactivated on the basis of MD simulations[35]. Furthermore, the width of the neck of the pore is consistent with the closed form of AtMSL1, where the pore gate is formed by a F323 ring located slightly lower[27] (Supplementary Fig. 4c). Nonetheless, heuristic predictions using the Channel Annotation Package (CHAP)[36] suggest that the residues lining the pore of FLYC1 and the presumed closed AtMSL1 do not create an energetic barrier that prevents water flow, in contrast with EcMscS in its closed state (Supplementary Fig. 5a–c). The lack of a hydrophobic barrier is caused by a widening of the pore and a reduction in local hydrophobicity at the level of V568 and F572 (corresponding to residues V319 and F323 in AtMSL1, and L105 and L109 in EcMscS, respectively).

We conducted all-atom molecular dynamics simulations to better analyze the conductance behavior of the pore conformation of FLYC1 captured in our cryo-EM structure, in particular the

role of F572 as a potential gate or selectivity filter. In AtMSL10, the homologous residue, F553, is important in maintaining the open state stability and controls channel conductance[25]. FLYC1 models with all protomers in the 'up' state was embedded in a pure 1-palmitoyl-2-oleoyl-sn-glycero-3-phosphocholine (POPC) membrane and solvated with water and 500 mM NaCl. We used a high salt concentration, coupled with an external electric field, to maximize the number of ion crossing events observed on a 100 ns timescale. Tension was not applied to the bilayer during the simulations. Analysis of the trajectories revealed the minimum radius along the pore averages at 2.8 Å (s.d. 0.3 Å, Fig. 2c), and that it presents no energy barrier to pore wetting. This confirms the heuristic prediction that F572 does not represent a closed hydrophobic gate in this conformation. Furthermore, this region also remains wetted throughout equilibrium simulations with no potential differences across the bilayer, suggesting the wetting behavior is not an artefact caused by the application of an external electric field (Supplementary Fig. 6). During these backbone-restrained simulations, the side chains of F572 underwent frequent rotameric switches from the starting *trans* rotamer ($\chi_1 \approx -180°$) to the *gauche* ($\chi_1 \approx -60°$) rotamer. This partly contributes to the variability of measured pore radius during the simulations (Supplementary Fig. 7). Next, we examined how permeable Cl⁻ ions travelled through the pore region in the presence of an external negative electric field. Over a total simulation time of 300 ns, 36 Cl⁻ complete efflux events were observed (Fig. 2d). The transit times of Cl⁻ ions from the entry through the side portal to the exit via the central pore varied greatly due to electrostatic interactions with residues in the cytoplasmic cage. A typical crossing event in simulation is shown in Supplementary Movie 1. Moreover, Cl⁻ remained hydrated when travelling through the narrowest constriction formed by the F572 residues (Fig. 2e). Based on the simulations, the

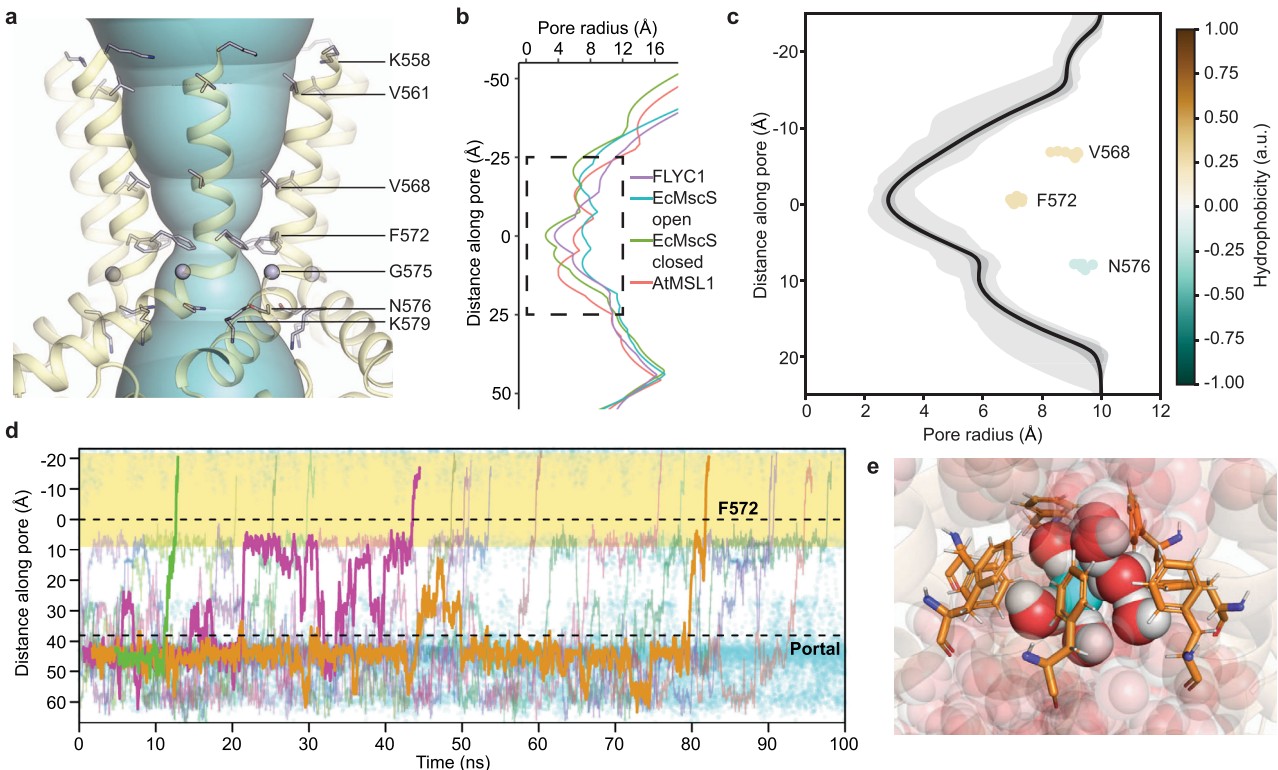

**Fig. 2 Pore of FLYC1. a** Depiction of central pore along the TM domain, with pore lining residues (shown as sticks, except for G575 which is shown as a sphere) labeled. **b** Pore profile of FLYC1, closed AtMSL1 and EcMscS in open (PDB: 5AJI) and closed states. Box represents area of (**c**). **c** Radius profile of FLYC1 pore (TM region) during a 100 ns protein backbone-restrained simulation. The dark gray region covers the maximum and minimum radii of the position, while the light gray region covers a region of mean radius ± 1 s.d. Positions of the pore-lining residues (center of mass) from the pore axis are labelled. **d** Trajectories of Cl⁻ traversing the channel completely during simulation in the presence of a −425 mV transmembrane potential difference, shown as their coordinates along the pore (z) axis. Each colored trace represents a different Cl⁻, while all Cl⁻ inside the channel is depicted in cyan. Three representative trajectories are highlighted in the foreground. The pore region lined by TM6a (from K558 to N576) is highlighted in yellow. The locations of the F572 ring and the cytoplasmic portal are also marked as dashed lines. One of our three independent simulations is shown for (**c** and **d**). **e** A representative snapshot of Cl⁻ ion crossing event through the F572 ring, during which dewetting behavior is not observed.

conductance of the model in the 'all-up' state can be estimated as ~50 pS (Supplementary Table 2). Electrophysiological measurements indicated a FLYC1 channel conductance of 164 pS in physiological recording solution that contains 130 mM NaCl, 5 mM KCl, 1 mM CaCl₂, and 1 mM MgCl₂, and conductance of 276 pS in 150 mM NaCl containing solution[8]. Therefore, the 'all-up' state does not correspond to a fully-conducting channel, and the central pore would have to dilate further to achieve a fully open state. During MD simulation we also observed Na⁺ ions (one in each 100 ns simulation) entering the cytoplasmic cage via the central pore, but a complete exit through the side portals were not observed at this timescale. The ratio of Cl⁻ to Na⁺ crossing events is broadly consistent with our previous reported $P_{Cl}/P_{Na}$ value of 9.8 ± 1.8 calculated on the basis of reversal potential in asymmetrical NaCl solution[8].

**Lysine residues in the side portals affect ion conduction.** Which parts of the structure contribute to ion selectivity and conduction properties of FLYC1? Charge reversal mutations of the only two pore-facing charged residues in TM6 (K579E and K558E) did not substantially alter pore properties[8], suggesting that other domains play a role. In bacterial MscS channels[37] and AtMSL1[28], the side portals of the cytoplasmic cage are determinants of ion selectivity and conduction. FLYC1 contains similar side portals at the intersubunit interface (Fig. 3a). To test the contribution of lysine residues that line the portal to ion permeation, we expressed K606

and K624 charge reversal mutations of FLYC1 constructs in HEK-P1KO cells and recorded stretch-activated currents[38,39]. Single and double mutants of K606E and K624E remained functional when stretch-activated currents were recorded under physiological conditions in the cell-attached patch clamp mode (Fig. 3b). Remarkably, all mutants exhibited reduced channel conductance when single channel currents were recorded from inside-out excised patches in symmetrical 150 mM NaCl solution (Fig. 3c). Independent glutamate substitutions at 606 and 624 resulted in fivefold and threefold reduction in channel conductance (WT: 270 ± 10 pS (N = 5); K606E: 54 ± 1 pS (N = 4); K624E: 156 ± 7 pS (N = 5)) (Fig. 3d). Notably, the double mutant further decreased channel conductance by twelve-fold (K606E, K624E: 22.6 ± 0.6 pS (N = 5), suggesting an additive effect (Fig. 3c, d). Furthermore, the Chloride to Sodium permeability ratio of the double mutant was reduced threefold ($P_{Cl}/P_{Na}$ = 3.2 ± 0.3, (N = 5)) relative to reported values for WT ($P_{Cl}/P_{Na}$ = 9.8)[8] (Supplementary Fig. 8). Together, these results conclusively indicate that the side portals of the cytoplasmic cage are integral to the ion permeation pathway.

The radius of each side portal was estimated to be ~2.1 Å (s.d. 0.3 Å), based on the simulations. While this was narrower than the constriction formed by F572 in the pore, this still did not present any energy barrier for wetting. Indeed, inspection of all Cl⁻ portal entry events across our simulations revealed that Cl⁻ remains hydrated during portal entry. In our MD simulations, Cl⁻ entered the side portal into the channel interacting first with K624 and

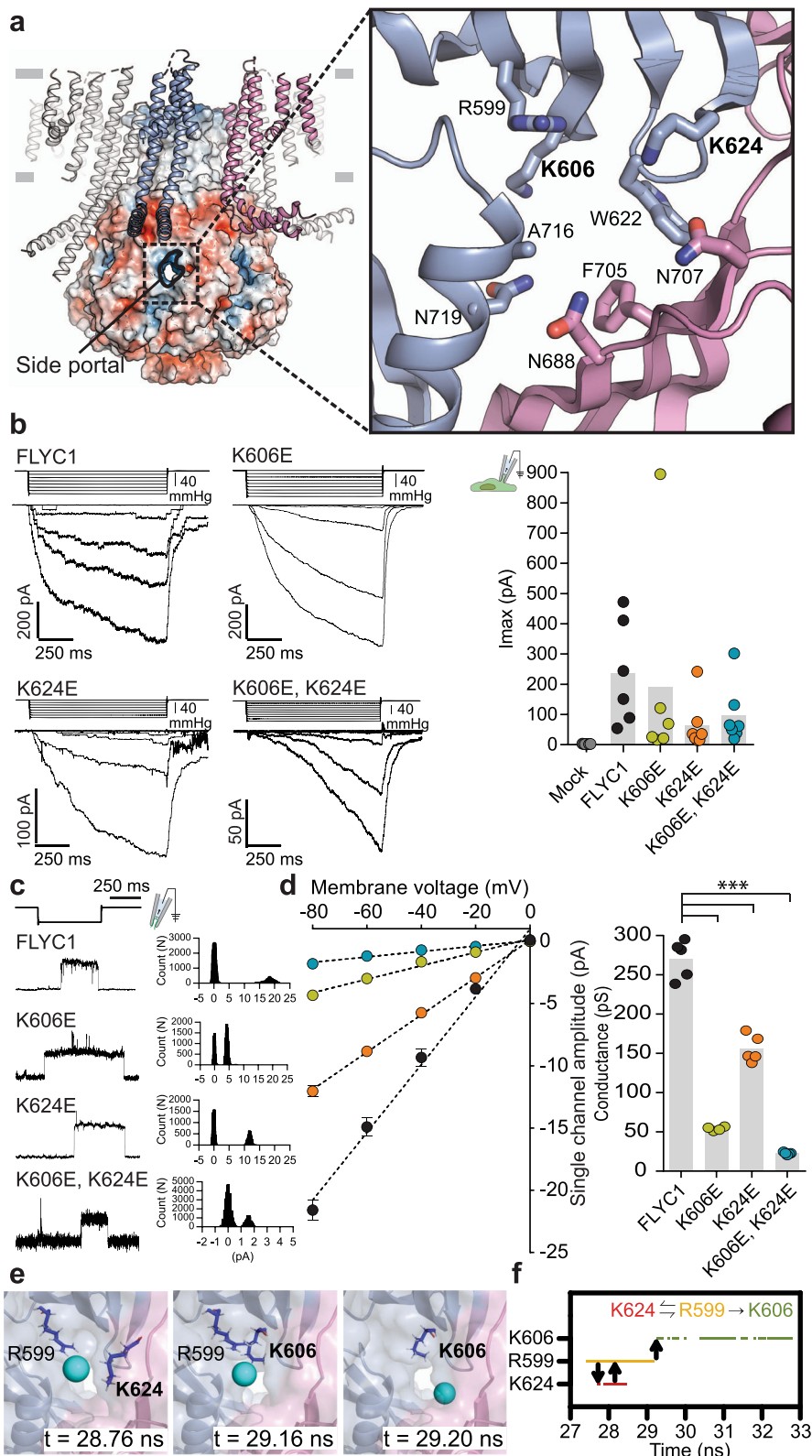

R599 (both outward-facing residues), then subsequently with K606 (inside the cytoplasmic cage) (Fig. 3e). The handover of Cl⁻ from R599 to K606 was unidirectional in all observed entry events (Fig. 3f). Taken together, these results suggest that indeed the side portals form part of the permeation pathway and can influence ion conduction properties of FLYC1, analogous to observations in MSL1 and EcMscS.

**Protomers of FLYC1 can be found in two distinct states.** While TM6 and the cytoplasmic cage, both close to the central axis, appear to be stabilized by interdomain and intersubunit contacts, the remainder of the FLYC1 molecule has minimal packing within and between subunits, allowing for significant conformational flexibility. To further resolve these areas, we used seven-fold symmetry expansion and focused classification to resolve the

**Fig. 3 Side portal of FLYC1. a** Cartoon and electrostatic surface representation of FLYC1 and cytoplasmic side portal between two subunits. Inset: expanded view of portal-lining residues as sticks. Residues selected for mutagenesis in bold. **b** Left, representative trace of stretch-activated currents recorded from WT or mutant FLYC1 expressing HEK-P1KO cells in cell-attached patch clamp configuration at −80mV membrane potential in response to Δ10 mmHg pipette pressure pulse. Stimulus trace illustrated above the current trace. Right, quantification of maximal current response from cells transfected with mock ($N = 7$), FLYC1 plasmid ($N = 6$), or FLYC1 plasmid with K606E ($N = 6$), K624E ($N = 7$), or K606E, K624E ($N = 7$) mutations. **c** Representative single channel traces in response to stretch from excised patches in symmetrical 150 mM NaCl at −80 mV from the indicated FLYC1 protein and their respective amplitude histograms. **d** Left, average I-V relationship of stretch-activated single channel currents from WT or mutant *FLYC1* transfected cells. I-V data from individual cells were fit with a linear regression curve and the slope was used to measure the conductance, plotted on the right; ***$p < 0.0001$, Holm–Sidak's multiple comparisons test relative to FLYC1. In panels **b** and **d**, individual cells are illustrated as scatter and mean is represented by grey bars. In left panel of **d** data points are mean ± S.E.M. $N =$ number of cells tested from different experimental days. **e** Representative snapshots of a Cl⁻ entering the channel via the side portal, interacting with basic residues in the vicinity. **f** An interaction scheme of the representative Cl⁻ entry event depicted in (**e**). The proposed ladder of interaction is consistent with all Cl⁻ entry events confirmed by inspection of each individual trajectory.

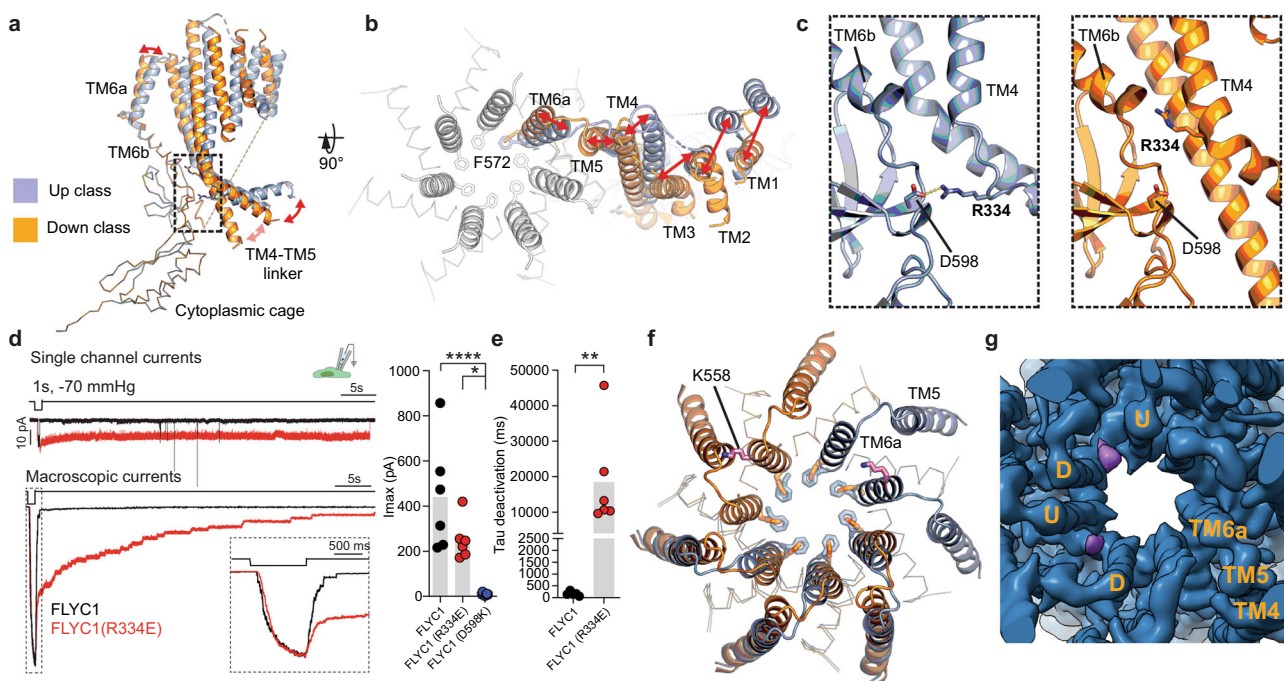

**Fig. 4 Conformational dynamics of FLYC1 protomer.** Side (**a**) and top (**b**) views of superposed up and down classes. Movement of helices denoted by red arrows. F572 shown as sticks. In top view, TM6a of all subunits is shown. **c** Expanded view of window in **a**, showing the presence of a salt bridge between R334 and D598 in the up (left) but not in the down class (right). **d** Left, representative stretch-activated currents from WT (black) or R334E mutant (red) FLYC1 in the cell-attached patch clamp mode in response to 1 s pulse of −70 mmHg pressure at −80 mV. For comparison, currents are normalized and overlaid. Top, single channel current; bottom, macroscopic current. Inset depicts enlarged section of the macroscopic trace to highlight that channel activation and inactivation are largely similar between WT and R334E. Right, quantification of maximal current response from cells transfected with *FLYC1* plasmid ($N = 6$), R334E mutant ($N = 6$), or D598K mutant ($N = 9$). ****$p = 0.000036$; * $p = 0.0104$, one-way ANOVA. **e** Mean deactivation kinetics of WT ($N = 6$) or mutant ($N = 6$) FLYC1 channel; ** $p = 0.0043$, Mann–Whitney test, two-tailed. In panel **d** and **e**, individual cells are illustrated as scatter and mean is represented by grey bars. $N =$ number of cells tested and pooled from different experimental days. **f** Superposition of hypothetical C7 'all-up' and 'all-down' classes. Two top left and two top right subunits show only down and up protomers, respectively. F572 ring is shown for all subunits in both conformations. The position and orientation of K558 are shown in pink for only two subunits. **g** First frame of first component of 3D variability analysis. Subunits are labeled up (U) or down (D) depending on their conformation. Lipid-like density is colored purple.

two distinct classes of the FLYC1 protomer we initially saw in our C1 reconstruction (Supplementary Fig. 1). A striking difference between the classes is the conformation of the cytosolic TM4-TM5 linker, which can be in an up or down conformation (Fig. 1a, b). In the up conformation, the TM4-TM5 linker is parallel to the membrane plane, while in the down conformation, the linker swivels towards the cytosol (Fig. 4a). Importantly, the conformational flexibility of the cytosolic domains is associated with movement of the TMD, suggesting that it is likely associated with channel gating. TM1-TM5 are displaced in a clockwise manner in the down state compared to the up state when viewed from the extracellular side (Fig. 4b). In the down state, a short six-residue coil (P96-S101) from the N-terminus interacts near the

side portals of the cytoplasmic vestibule (Supplementary Fig. 9a). This peptide is partially conserved between MSL8, MSL9, and MSL10 sequences (Supplementary Fig. 3). Symmetry expansion followed by variability analysis in cryoSPARC[40] confirmed the presence of density for this short peptide only in the down conformation (Supplementary Fig. 9b and Supplementary Movie 2). Low-pass filtering of the unsharpened C1 map to 6 Å revealed a diffuse density between said peptide and the TM4–TM5 linker, pointing to a potential interaction between this domain and the N-terminus (Supplementary Fig. 9c). Although such peptide is not detected in the up class, in this instance, the low-pass filtered map showed the TM4–TM5 linker and the cytosolic N-terminus of TM1 bridged by a similarly diffuse

density. Thus, the dynamic TM4–TM5 linker may interact with the cytosolic-facing side TMD periphery or the central cytoplasmic cage, depending on whether the protomer is in an up or down state.

Based on the composite '6 up 1 down' model, we carried out additional MD simulations in the presence of negative membrane potentials of −450, −225, and −112.5 mV. The pore radius at the start is ~10% wider for the composite model at 3.1 Å radius, compared to the 'all-up' model. We also note that the pore constriction geometry across these three simulations is more well-preserved when the composite model was used (Supplementary Fig. 7). The wider pore constriction is accompanied by an increased number of chloride ion crossing events over the same simulation timescale, with a conductance estimation much closer to the electrophysiological measurements of the stretch-activated currents (Supplementary Table 2). The conformational flexibility of the TM4–TM5 linker was further studied in a separate 100 ns simulation with no restraints applied (Supplementary Fig. 10). Without restraints, subunits originally in the 'up' state are able to switch downwards as the modeled linker does not contact the lipid bilayer and other subunits (though a significant portion of the linker of about 150 residues are unmodeled and could mediate inter-domain/subunit interactions), resulting in a breakdown of local symmetry.

In the up state, the cytosolic extension of TM4 interacts with the β-domain of the cytoplasmic cage through a salt bridge between R334 and D598 (2.9 Å distance) (Fig. 4c). In the down state, R334 shifts upward and does not contact D598. Interestingly, in EcMscS, a loosely analogous interaction between the cytoplasmic end of TM1–TM2 (corresponding to TM4-5 in FLYC1) with TM3b has been observed to affect gating kinetics[41,42]. Given that the cytosolic extension of the flanking TM helices is a topologically unique feature of FLYC1 and the specificity of the R334–D598 interaction to the up state, we hypothesized that it might play a role in channel gating. To test this, we disrupted the R334–D598 salt bridge by either mutating the arginine to glutamate or mutating the aspartate to lysine and determined its effect on channel kinetics. D598K mutation rendered the channel non-functional, but remarkably, stretch-activated currents from R334E mutant exhibited a drastic increase in open-dwell time in both macroscopic and single-channel currents (Fig. 4d). Effectively, the deactivation time constant of the channel was approximately 40-fold slower than WT (WT: 0.477 ± 0.285 s ($N = 6$); R334E: 18.5 ± 5.2 s ($N = 6$)) (Fig. 4e). A similar phenotype has been observed in AtMSL1, where A320V substitution in the TMD resulted in spontaneous channel opening and longer open dwell times[27]. However, the molecular basis is most likely different; introduction of V320 in AtMSL1 is expected to disturb packing of the pore helix (TM5)[27], whereas R334E in FLYC1 disrupts a conformation-specific interaction between cytoplasmic domains. WT FLYC1 does not exhibit a stretch-dependent inactivation characteristic of EcMscS, and the R334E mutation does not alter this behavior[8] (Fig. 4d). It is likely that R334E mutation stabilizes the channel in an open state, suggesting that a dynamic interaction between the TM4–TM5 linker and the cytoplasmic cage is indeed a crucial step in FLYC1 channel gating.

TM6a hinges away from the pore center in the down state, which displaces F572 outward (Fig. 4b). In E. coli MscS, a hinging motion in TM3 (TM6 in FLYC1) is thought to underlie gating[14]. Therefore, the distinct TM6 conformations in up and down states could represent different functional states. Hypothetical C7 models of FLYC1 with all protomers in up or down state were superposed to evaluate changes in the pore. The radius of the pore constriction in the 'all-down' state increased by ~1.3 Å and resembled the width of subconducting state of EcMscS[30]

(Supplementary Fig. 5d). However, such a heptamer presents steric clashes between TM5 and TM6a of the neighboring subunit, with K558 causing the most significant clashes (Fig. 4f). Hence, if an all-down state exists, additional re-arrangement of the TMs would be required, which would likely alter the pore further.

3D EM variability analysis displayed a lipid-like density at the top of the pore, which differs from the 'hook' and pore lipids of EcMscS, as it is not on the outer side of TM6a or in the middle of the pore, respectively (Supplementary Fig. 11)[30–32], but rather it intercalates in-between adjacent TM6a helices. This density is associated with the down class, but it only appears when the adjacent subunit (clockwise when viewed from extracellular side) is positioned far enough to create a side opening to the membrane (Fig. 4g, and Supplementary Movie 2). In our dataset, this gap is created when the preceding subunit is in the up state. We did not observe two contiguous down protomers in our data, so the presence of the density in such case cannot be determined. The presence of this lipid-like density suggests that the down state causes a separation of the protomers that can be filled by lipids. Whether lipids can access the pore, as suggested for EcMscS[30,31], is not determined.

## Discussion
Our work presents the structure of FLYC1, a MscS homolog with a conserved central core structure and flanking domains with novel architecture. Our structure captured a TM domain conformation that is distinct from those previously observed for MscS/MSL1. Our MD simulations support the idea of passage of chloride in an 'all-up' state at −425 mV and a '6 up 1 down' state at −450 mV, with the latter exhibiting an estimated conductance closer to electrophysiology measurements. However, electrophysiology recordings suggest that although FLYC1 favors chloride flux, it is also permeable to sodium ions[8], which, when fully hydrated, are larger than hydrated chloride[43]. We did not observe any complete $Na^+$ permeation events within the timescale of our simulations, suggesting that our structure of FLYC1 does not represent a fully open conformation. Furthermore, a small degree of constriction of the FLYC1 pore, either due to movement of the helices and/or intrusion of lipid tails, could result in functional closure of the pore due to hydrophobic gating. Hence, the structure of FLYC1 obtained in this study likely represents a near-closed conformation, or perhaps a subconducting state as observed in AtMSL10-expressing oocytes[24]. In spite of this, at the single protomer level, the observed FLYC1 conformation is most similar to open EcMscS[30]. The marked differences between the near-closed structure of FLYC1 and other closed-state structures of MscS/MSL channels point to a distinct gating mechanism for FLYC1.

Previous studies of MscS/MSL family channels have focused on the role of the TM domains in mechanosensitive gating[12,14,27,31,32,42]. Our structure-function data demonstrates that the unique cytoplasmic flanking regions of FLYC1 play a significant role, with the partially structured cytosolic TM4–TM5 linkers pivoting between up and down states. The striking 40-fold delay in deactivation kinetics upon disruption of the up state-specific R334–D598 interaction suggests that the formation of this salt bridge in the WT protein is associated with channel closure and that losing this interaction is important for the stability of the open state. We speculate the R334E mutant channel remains in an open conformation for a longer duration because the mutation increases the energetic barrier for the transition from open to the closed state. This could be due to an additional interaction in the open state by the introduced E334 residue or the R334–D598 interaction favoring the closed state. The non-functionality of the

complementary mutation D598K could support the first hypothesis, yet we are unable to exclude trafficking or folding issues of this mutant at this time. Similar issues have been observed for overexpression of AtMSL10 in HEK-P1KO cells, where no currents were observed despite these channels inducing mechanosensitive currents when expressed in oocytes, possibly due to trafficking[8,24]. We, therefore, predict that the fully open form of FLYC1 would involve down or other 'non-up' states of the TM4–TM5 linker, and that channel gating transitions would involve concerted motions of the cytoplasmic flanking domains and TM3–TM6 helices. Likewise, the stability of the closed state in the WT where this salt bridge is, presumably, formed suggest that the up conformation may be favored in the resting state. Our observation that the channel adopts asymmetric assemblies with protomers adopting either up or down states is likely to add further complexity to the FLYC1 gating mechanism.

Sequence homology suggests the TM4–TM5 linker region and the cytoplasmic N-terminus will have similar structure in MSL8, MSL9, and MSL10 channels. In addition to the TM4–TM5 linker, we noted that the cytoplasmic N-terminus of FLYC1 interacts dynamically with the cytoplasmic cage. Interestingly, the interaction between the N and C-terminus of AtMSL10 has been proposed to induce cell death, for instance in response to swelling in seedlings[44,45]. Whether this proposed interaction in FLYC1 corresponds to the one observed in FLYC1 remains unclear.

The unique features of FLYC1 described here may underlie its specialized function as a mechanosensor in Venus flytrap. However, it is worth noting here that two other genes have also been identified as potential candidates modulating trap closure, DmFLYC2 and DmOSCA[8]. Nonetheless, our structural and functional characterizations of FLYC1 highlight potential conformational transitions in the channel's gating scheme, and will inform future investigations into the mechanisms of mechanosensation and gating in the MscS superfamily.

## Methods

**Construct, expression and purification.** *Flycatcher1* was cloned into a pEG BacMam vector[46] as a C-terminal GFP fusion with an intervening short flexible linker followed by PreScission protease cleavage site (Gly-Ser-Gly-Ser-Leu-Glu-Val-Leu-Phe-Gly-Pro). The distal C-terminus of this construct also contained a streptavidin-biding peptide tag (Trp-Ser-His-Pro-Gln-Phe-Glu-Lys). We denote this construct FLYC1-pp-GFP.

FLYC1-pp-GFP was expressed in Human Embryonic Kidney (HEK) 293F cells by baculovirus transduction[46]. Baculovirus was produced in Sf9 cells. 10% v/v baculovirus was added to HEK 293F cells (grown in suspension at 37 °C and 8% $CO_2$) when they reached a density of ~2 × 10⁶/mL. After 8 h, 10 mM sodium butyrate was added to the cells and the temperature was shifted to 30 °C, and incubated for an additional two days. All purification steps were conducted at 4 °C or on ice. Cells were pelleted by centrifugation, washed with PBS, and resuspended in H buffer (25 mM HEPES pH 8.0, 150 mM NaCl, 2 µg/mL leupeptin, 2 µg/mL aprotinin, 2 µM pepstatin A, 1 mM phenylmethylsulfonyl fluoride, and 2 mM dithiothreitol). Cells were lysed by sonication and cell debris were pelleted by centrifugation at 4,259 × g. Membranes were isolated by centrifuging the supernatant at 125,171 × g for one hour, then mechanically homogenized in Dounce homogenizer in buffer H. Protein was extracted by stirring the homogenized membranes in buffer H supplemented with 0.9% glyco-diosgenin (GDN) and 0.1% C12E9 for one hour, then clarified by centrifugation and incubated with 2 mL of home-made GFP nanobody linked Sepharose resin[47,48] for 1.5 h. The resin was collected in a gravity column and washed with 50 mL of wash buffer (25 mM HEPES pH 8.0, 300 mM NaCl, 2 µg/mL leupeptin, 2 µg/mL aprotinin, 2 µM pepstatin A, 1 mM phenylmethylsulfonyl fluoride, 2 mM dithiothreitol, 0.04% GDN, 0.01% C12E9) then resuspended in SEC buffer (25 mM HEPES pH 8.0, 150 mM NaCl, 0.4 µg/mL leupeptin, 0.4 µg/mL aprotinin, 0.4 µM pepstatin A, 0.2 mM phenylmethylsulfonyl fluoride, 0.4 mM dithiothreitol, 0.04% GDN). Protein was cleaved off resin by addition of ~300 µg PreScission protease to the slurry followed by incubation for 5 h at 4 °C. Flowthrough was concentrated in a 100 kDa MWCO Amicon centrifugal filter and injected into Superose 6 increase column equilibrated to SEC buffer. Peak fractions corresponding to heptameric FLYC1 were pooled and concentrated to 7 mg/mL in a 100 kDa MWCO Amicon centrifugal filter for cryo-EM.

Grid preparation was carried out in a Vitrobot Mark IV (ThermoFisher) operated at 10 °C and 100% humidity. 3.5 µL of concentrated FLYC1 protein

sample was applied to a freshly plasma cleaned UltrAuFoil grid (1.2/1.3 hole size/ spacing, 300 mesh) and blotted for 3 s, then plunge frozen in liquid ethane cooled by liquid nitrogen. The grids were stored in liquid nitrogen until data collection.

**Cryo-EM data collection and processing.** Two datasets were collected using a Titan Krios (ThermoFisher) operating at 300 kV with a K2 Summit direct electron detector (Gatan) with a pixel size of 1.03 Å. Leginon was used for automated data collection[49]. Between the two datasets, 5159 movies were collected with a total accumulated dose of ~50 electrons per Å² (42 or 43 frames per movie) with a defocus range of −2.2 to −0.7 µm. Frame alignment and dose weighting were carried out using the RELION implementation of MotionCor2[50,51]. Each dataset was treated as a separate optics group. Good micrographs were selected using MicAssess[52] (0.05 threshold) and imported into cryoSPARCv2[53]. CTF values were estimated using GCTF[54]. Blob picking from a set of 200 micrographs lead to the generation of an ab initio volume followed by 1 round of 3D refinement. Unless specified, all 3D refinements done in cryoSPARC were non-uniform refinements ("Legacy" in version 3) with C1 symmetry. Templates for picking the whole dataset were created based on the initial map. 1,077,949 particles were extracted with a box size of 280 pixels and subjected to 2D classification. Particles in best classes were subjected to heterogeneous refinement with 6 classes. 364,333 particles corresponding to 1 class were further refined imposing C7 symmetry, followed by global and local CTF refinement. After an additional C7 refinement, particles were classified (C1, 3 classes) and best class was refined one last round. This clean particle stack was later used for variability analysis (see below). The 255,359 particles of this stack were imported into RELION-3.1[55]. Particles were re-extracted based on the output of CTFFIND4[56] and then 3D refined without symmetry. From this point on, all 3D refinements were done using SIDESPLITTER[57], with local angular searches only and C1 symmetry, unless otherwise stated. A second 3D refinement was followed by three consecutive rounds of CTF refinement[58], followed by 3D auto-refine, Bayesian polishing[51], and 3D auto-refine. From the first Bayesian polishing step onward, processing continued with 129,933 particles corresponding to only one of the two datasets. The map at the end of the three rounds was post-processed with DeepEMhancer[59] and the resulting map is referred to as the 'C1 map'. In all instances DeepEMhancer was used, the 'tightTarget' model was employed for post-processing with the two half-maps as input. Separately, the same stack was subjected to a 3D refinement with C7 symmetry followed by symmetry expansion using RELION, where 7 total copies of the particle stack were created and each copy rotated by ~51° (360° divided by 7) relative to the previous copy[60]. The micelle around the protein was removed via particle subtraction. A mask containing one protomer, but wide enough to include TM6a and small portion of cytoplasmic domain of adjacent subunits, was used for focused 3D classification without alignment. From the resulting six classes, one class of 145,362 particles corresponded to the down conformation of the protomer, while two classes, with 651,815 particles between them, were assigned to the up conformation. The up and down conformations were processed separately after this point. Each particles set was reverted to the original particles before subtraction subjected to 2–3 rounds of 3D auto-refine, revealing density for the outer TMs. Both maps were post-processed with DeepEMhancer, producing the up and down-focused maps. For each of the focused maps, the protomer with the most TM helices visible was extracted and aligned to the corresponding subunit in the C1 map (6 up and 1 down protomers) in Chimera and then merged (vop maximum function) to create the composite map. The unsharpened C1, up focused, and down focused maps (before DeepEMhancer) were used to calculate the FSC and local resolution in RELION.

**Variability analysis.** In cryoSPARCv3, a clean stack of 255,539 particle (same stack imported into RELION for analysis described above) was symmetry expanded (C7). The expanded stack was used as input for 3D variability analysis[40] with four nodes using low and high-pass filters of 6 and 20 Å, respectively. A volume series of 20 frames was obtained by using the 3D variability display job in simple mode. All other parameters not specified were used with default values. Movie of variability analysis output was made with UCSF Chimera[61].

**Model building and refinement.** The molecular model for FLYC1 was built de novo using the up class in Coot v0.9[62,63] and PDB 2OAU as a guide. The model was refined using real space refinement in Coot and Phenix[64,65]. The model for the down class was then built using the model for the up class as a starting point. Six protomers of the up class and one protomer of the down class were fitted into the appropriate density of the composite map using UCSF Chimera[61]. The model was then subjected to iterative rounds of building in Coot and refinement in Phenix and Rosetta[66]. ISOLDE[67] was used at different stages of the process to identify regions that were poorly modelled. Restraints for palmitic acid were generated with eLBOW[68] from its SMILES code. The final model of FLYC1 includes residues 96–101, 258–285, 294–354, 503–751 for the down class and residues 264–285, 290–354, 507–751 for the up class. Residues 186–204 and 222–244 in the down class and residues 186–203 and 222–242 in the up class (TM1 and TM2) are included in the model but have been assigned as poly-Ala due to low resolution of these regions of the map. The numbering of these residues was assigned based on TOPCONS[69] predictions for TM1-2. Model was validated using MolProbity[70],

EMRinger[71], and phenix mtriage[72] to calculate the map to model FSC. The final model was validated against the composite map, while the up and down protomers were validated against their respective focused map. For validation of the up class, we used the protomer positioned adjacent to the subunit in down conformation in an anticlockwise direction when viewed from the extracellular side. For completeness, a model excluding TM1–TM3 of all protomers was validated against the C1 map. The values of the validation are reported in Supplementary Table 1. Pore profiles were calculated with HOLE[73] and heuristic predictions of pore hydration on the transmembrane region of FLYC1 and homologs were calculated with CHAP[36,74]. Structure figures were generated in PyMOL[75], UCSF ChimeraX[76], and UCSF Chimera[61]. Amino acid sequence alignment was done with Clustal omega[77] and represented with ESPript3[78].

Deposited models of FLYC1 homologs for our structural analysis were obtained from the PDB[79]. The PDB IDs are the following: for EcMscS closed state is 2OAU[12], EcMscS open state 5AJI[15], EcMscS desensitized state 6VYM[30], EcMscS subconducting state 6VYL[30], EcYnaI closed state 6ZYD[32], EcYnaI open state 6ZYE[32], EcYbiO 7A46[32], AtMSL1 closed state 6VXM[27], and AtMSL1 open state 6VXN[27].

**Cell culture and transient transfection**. PIEZO1-knockout HEK 293T (HEK-P1KO) were used for all heterologous expression experiments. HEK-P1KO cells were generated using CRISPR–Cas9 nuclease genome editing technique as described previously[80], and were negative for mycoplasma contamination. Cells were grown in Dulbecco's Modified Eagle Medium (DMEM) containing 4.5 mg.ml$^{-1}$ glucose, 10% fetal bovine serum, 50 units.ml$^{-1}$ penicillin and 50 μg.ml$^{-1}$ streptomycin. Cells were plated onto 12 mm round glass poly-D-lysine coated coverslips placed in 24-well plates and transfected using lipofectamine 2000 (Invitrogen) according to the manufacturer's instruction. All plasmids were transfected at a concentration of 700 ng.ml$^{-1}$. Cells were recorded from 24 to 48 h after transfection. Mutations in FLYC1 were introduced using Q5 site-directed mutagenesis kit (New England Biolabs).

**Electrophysiology**. Patch-clamp experiments in cells were performed in standard cell-attached, or excised patch (inside-out) mode using Axopatch 200B amplifier (Axon Instruments). Currents were sampled at 20 kHz and filtered at 2 kHz or 10 kHz. Leak currents before mechanical stimulations were subtracted off-line from the current traces. Voltages were not corrected for a liquid junction potential (LJP) except for ion selectivity experiments. All experiments were done at room temperature and data was analyzed using Clampex 10.6 and GraphPad Prism.

Solutions: For cell-attached patch clamp recordings, external solution used to zero the membrane potential consisted of (in mM) 140 KCl, 1 MgCl$_2$, 10 glucose, and 10 HEPES (pH 7.3 with KOH). Recording pipettes were of 1–3 MΩ resistance when filled with standard solution composed of (in mM) 130 NaCl, 5 KCl, 1 CaCl$_2$, 1 MgCl$_2$, 10 TEA-Cl, and 10 HEPES (pH 7.3 with NaOH) or 150 NaCl and 10 HEPES (pH 7.3 with NaOH). Single channel currents were recorded in excised inside-out patch configuration in external and pipette solution containing (in mM): 150 NaCl and 10 HEPES (pH 7.3 with NaOH). Ion selectivity experiments were performed in inside-out patch configurations. $P_{Cl}/P_{Na}$ was measured in extracellular solution composed of (in mM) 150 NaCl and 10 HEPES (pH 7.3 with NaOH) and intracellular solution consisted of (in mM) 30 NaCl, 10 HEPES, and 225 sucrose (pH 7.3 with NaOH)[8].

Permeability ratio measurements: Reversal potential for each cell in the mentioned solution was determined by interpolation of the respective current–voltage data. Permeability ratios were calculated by using the following Goldman–Hodgkin–Katz equations:

$$E_{rev} = \frac{RT}{F} \ln \frac{P_{Na}[Na]o + P_{Cl}[Cl]i}{P_{Na}[Na]i + P_{Cl}[Cl]o} \tag{1}$$

Mechanical stimulation: Macroscopic stretch-activated currents were recorded in the cell-attached or excised, inside-out patch clamp configuration. Membrane patches were stimulated with 1 s negative pulses through the recording electrode using Clampex controlled pressure clamp HSPC-1 device (ALA-scientific), with inter-sweep duration of 1 min. For single-channel currents, because amplitude is independent of the pressure intensity the most optimal pressure stimulation was used to elicit responses that allowed single-channel amplitude measurements. These stimulation values were largely dependent on the number of channels in a given patch of the recording cell. Single-channel amplitude at a given potential was measured from trace histograms of 2–4 repeated recordings. Histograms were fitted with Gaussian equations using Clampfit 10.6 software. Single-channel slope conductance for each individual cell was calculated from linear regression curve fit to single-channel I–V plots.

**Constant electric field all-atom molecular dynamics simulations**. MD simulations and subsequent analyses were performed using the 'all-up' and the composite models ('6 up 1 down') of FLYC1. The coordinates of FLYC1 'all-up' were first converted to a coarse-grained representation (MARTINI 2.2 force field[81]) using MemProtMD[82], then embedded in a band of randomly oriented POPC (1-palmitoyl-2-oleoyl-sn-glycero-3-phosphocholine) molecules, and solvated on both sides with water and 0.5 M NaCl. A coarse-grained simulation of 100-ns was performed, with the protein backbone beads position restrained with a force constant of 1000 kJ mol$^{-1}$ nm$^{-2}$ for lipid self-assembly. The output frame was converted to an atomistic representation (OPLS all-atom protein force field with united-atom lipids[83]) using CG2AT-Align[84], then resolvated with TIP4P/2005 water[85] and 0.5 M NaCl. Triplicates of 100 ns production runs with a 2 fs timestep were then performed. To preserve the experimentally determined conformation but allowing for side chain flexibility, simulations were performed in the presence of position restraints of protein backbone atoms with a force constant of 1000 kJ mol$^{-1}$ nm$^{-2}$. One additional simulation involving the composite model was simulated without restraints to observe the behavior of the TM4-TM5 linker. To drive chloride ions passage from the cytosolic side to the extracellular side, an external uniform electric field of −6.25 to −25 mV nm$^{-1}$ (see Supplementary Table 2) was applied in the direction normal to the membrane, which corresponds to a maximum transmembrane potential difference of −450 mV[86]. Simulations were performed as NPT ensembles held at 1 bar and 310 K, maintained with a semi-isotropic Parrinello–Rahman barostat[87] (coupling constant $\tau_P = 1$ ps) and a velocity-rescaling thermostat[88] (coupling constant $\tau_T = 0.1$ ps). Covalent bonds were constrained through the LINCS algorithm;[89] electrostatics were modelled with a smooth particle mesh Ewald method;[90] and van der Waals interactions were modelled using a Verlet cut-off scheme. All simulations were performed using GROMAC 5.1.2[91].

**Pore pathway calculations and trajectory analyses**. The radius of the pore in the transmembrane region throughout the simulations was calculated using CHAP[36,74] excluding the first 10 ns in each of the 100 ns simulations, sampled every 200 ps. Initial probe position was set at the center of the F572 ring. The reported value of the pore minimum radius was the average across the triplicates. A similar approach was used to calculate the dimension of the side portals during the simulations. For each of the seven portals in a 100 ns run, a CHAP path-finding run was performed as above, with the initial probe position set at the center of each portal (defined as the geometric center of W622, F715, N719, and F705 of the adjacent subunit) and the channel direction vector set as towards the seven-fold axis along the plane of the membrane. The reported value of the portal minimum radius was the average across the seven portals. Cl$^-$ inside the channel were selected based on a set of distance-based constraints to the seven-fold axis of the channel. Cl$^-$ crossing events were counted as those ions that entered the side portal and exited the channel completely via the pore during the simulations. The conductance of FLYC1 in simulations was estimated by $G = I/V = Q/(t \times V)$.

**Reporting summary**. Further information on research design is available in the Nature Research Reporting Summary linked to this article.

## Data availability

The composite map for FLYC1 has been deposited to the Electron Microscopy Data Bank (EMDB) under accession code EMD-24186. Corresponding atomic coordinates of FLYC1 have been deposited to the PDB under IDs 7N5D. C1 symmetry map and focused maps in down and up conformation, along with atomic coordinates, used to generate the composite map have been deposited to the EMDB under codes EMD-24187, EMD-24188, and EMD-24189, and PDB under IDs 7N5E, 7N5F, and 7N5G, respectively. Frames have been deposited in EMPIAR under code EMPIAR-10740. Source data are provided with this paper.

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

## Acknowledgements
We thank W. Anderson for managing the electron microscopy facility at Scripps Research, H. Turner for helping with data collection, and C. Bowman, L. Dong, and J.C. Ducom for assistance with computation. We acknowledge members of the Ward and Patapoutian labs for helpful advice. This work was supported by NIH grant R01 HL143297 and a Ray Thomas Edwards Foundation grant to A.B.W., and NIH grant R01 HL143297 to A.P. Work in M.S.P.S.'s lab is supported by Wellcome (grant 208361/Z/17/Z), BBSRC (grants BB/N000145/1 and BB/R00126X/1), and EPSRC (grant EP/R004722/1). K.S. was a postdoctoral fellow of the Jane Coffin Childs Memorial Fund for Medical Research. C.C.A.T. is supported by the Skaggs-Oxford Scholarship and the Croucher Foundation. A.P. is an investigator of the Howard Hughes Medical Institute. Molecular graphics and analyses performed with UCSF Chimera and UCSF ChimeraX, developed by the Resource for Biocomputing, Visualization, and Informatics at the University of California, San Francisco, with support from National Institutes of Health R01-GM129325 and P41-GM103311, and the Office of Cyber Infrastructure and Computational Biology, National Institute of Allergy and Infectious Diseases.

## Author contributions
K.S. and W.H.L. cloned expression constructs and mutants. K.S. prepared protein samples and acquired cryo-EM data. S.J.C. and K.S. processed data and built and refined the atomic structures. S.E.M. carried out electrophysiology experiments and analyzed data. C.C.A.T. performed MD simulations under the supervision of M.S.P.S. A.B.W., and A.P. supervised the project. S.J.C., S.E.M., and K.S. drafted the manuscript, which was edited and finalized with contributions from all authors.

## Competing interests
The authors declare no competing interests.
