## [Peer Review File · Nature Communications]

Structural Insights into the Venus flytrap Mechanosensitive Ion Channel Flycatcher1REVIEWER COMMENTS

Reviewer #1 (Remarks to the Author):

This study by Jojoa-Cruz, Saotome et al. presents the first structures and associated molecular dynamics and patch-clamp recordings of the mechanosensitive ion channel Flycatcher1 from the Venus flytrap. FLYC1 is related to bacterial MscS and plant MSLs, but has different functional properties (including a higher Cl⁻ selectivity, lower single channel conductance, and lower mechanical threshold for activation) that might be accounted for by unique structural elements. The structures show a generally conserved architecture with new features including additional transmembrane helices and a unique cytoplasmic TM4-5 linker that adopts “up” or “down” conformations in different protomers within the heptameric channel. Molecular dynamics suggests the structure is weakly conductive, perhaps corresponding to a transition or subconductance state in the channel gating scheme. Residues in the cytoplasmic cage side portals are shown to be important for conductance and the TM4-5 up conformation is shown to stabilize a closed state. Overall, this is a very interesting and well executed study of this newly identified channel that sets the stage for future studies of gating and determinants of specific properties. I have only minor comments and suggestions for the authors consideration.

1. Is there a reason the authors chose to use an “all-up” structure for molecular dynamics rather than the “6 up 1 down” composite structure? As far as I understand, the 6 up 1 down composite was judged to best represent observed particle conformations in the C1 reconstruction. Perhaps MD on the composite could be performed and compared or connections between the MD and structural data qualified to reflect the difference (for example in the discussion “We did not observe any complete Na⁺ permeation events in our simulations at this timescale, suggesting that our structure of FLYC1 does not represent a fully open conformation”).

2. Is there experimental evidence for subconductance states that could correspond to that observed by MD? It is not obvious (to me) from the histogram in 3C.

3. It is striking that the pore diameters of FLYC1 and closed EcMscS (and AtMSL1?) are similar and both have phenylalanine ring constrictions, but only EcMscS is found computationally to have a hydrophobic barrier. Could the authors elaborate on chemical differences in the pore that could account for this? Perhaps labeling the residues or positions represented by boxes in S5 would make this easier to interpret?

4. Do the charge swap mutations K606 and K624 shown to reduce conductance also alter channel selectivity? If not, can the authors speculate on how Cl⁻ selectivity is achieved?

5. Is there any evidence the additional TM helices in FLYC1 contribute to functional differences between FLYC1 and EcMscS/AtMCL1?

6. Page 7: Is the question mark in the section title “A phenylalanine ring forms the pore gate of FLYC1?” intentional?

7. Page 7: “van der Waal” should be van der Waals

8. Figure 3b is missing a time scale bar.

Reviewer #2 (Remarks to the Author):

Jojoa-Cruz et al. present an important biophysical study that may reveal the fascinating molecular mechano-sensation mechanism of Venus flytraps, by combining structural biology, computational biology, and electrophysiology approaches. Although the overall structure of the MS ion channel Flycatcher1 is similar to previously solved homologous proteins such as MscS and AtMSL1, a new structural feature of the TM4-TM5 linker was observed, which may be related to a distinct gating mechanism. The MD simulations and electrophysiology experiments significantly strengthened the manuscript, providing atomistic, dynamic, and quantitative information. The manuscript is well written, clear and concise.

I have several questions/suggestions for the authors to address:

1) Can the TM4-TM5 linker reach the side portal in the 'down' conformation? Why does the 'down' conformation (or disruption of an up conformation-specific interaction) stabilize the open state? The authors ought to give some discussions on the likely underlying reason at least.

2) The mutation D598K led to a non-functional channel, why? Since the authors state that breaking the salt bridge between D598 and R334 should stabilize the open state / slow down the deactivation, as evidenced by the mutant R334E, one may expect that D598K would have a similar effect. The "unexpected" non-functional mutation D598K should be clarified.

3) The Y axes in Fig 2b and 2c are in the reverse directions, which makes it a bit difficult to understand the location in the pore, especially when Fig 2a-c are put together. I suggest the authors use the same y-axis direction and range for Fig 2b and 2c.

4) I would like to see a video showing a complete CL- permeation event in the MD simulations, which would be appealing to the audience too.

5) How dynamic is the TM4-TM5 linker in the MD simulations? How is the dynamics of the TM4-TM5 linker affected by the mutation R334E/D598K? These MD simulations/analyses may be informative for understanding the role of the TM4-TM5 linker.

Reviewer #3 (Remarks to the Author):

The manuscript 'Structural Insights into the Venus flytrap Mechanosensitive Ion Channel Flycatcher1' by Jojoa-Cruz and coworkers provides the first data on the structure and mechanism of this intricate version of an MSL-like channel that helps carnivorous plants detect prey. The cryo-EM structural information on the mostly resolved central core domains is extended by MD simulations, which estimate the dynamics and conductance of the pore in one of the conformations (up-state). The structure and simulations give important functional predictions regarding the ionic selectivity determinants and open-state stabilizing interactions. These predictions were supported by mutagenesis and patch-clamp experiments.

In the cryo-EM section, the authors judiciously applied the seven-fold symmetry constraints only to certain 2D classes at selected stages of analysis and treated others as C1 objects which allowed them to discern asymmetric conformations with 'up' and 'down' positions of the TM4-TM5 cytoplasmic linker in different protomers. This advanced approach paid off in identifying the minor 'down' component, which was functionally designated as a step toward opening. The technical aspects of this work raise no questions apart from my request to better explicate the concept of symmetry expansion. This methodology is distinct from most of the previous studies of MscS-like channels where C7 symmetry was applied by default.

My major reservation concerns the assignment of FLYC1 in the up-state as a subconductive conformation. The authors state that, based on the heuristic assessment, the gate of the channel has no tendency to dewet. It would be important to show a water density plot along the z axis as a function of time at zero voltage. It might be also good to justify the choice of TIP4 water in the Methods.

Fig 2a depicts some sort of water-accessible cross-section along the pore and only selected sidechains are shown (with no volume). Because TM6 helices in FLYC1 are packed less tightly than TM3s in EcMscS, it is expected that more water will be in the interhelical spaces and more polar atoms exposed to the pore water. For this reason, the authors should consider presenting the inner surface of a vertical pore cross-section colored by atom polarity.

The energy of a fully solvated Cl⁻ ion was not assessed along the path and only the number of ions passing the pore per ns was counted. The voltage of -425 mV is a concern since non-linear effects might be present and dielectric water attraction into the region of high electric field (pore constriction) is expected. What is the Cl⁻ permeation rate at 100 and 200 mV? In their 2019 PNAS paper Sansom and colleagues show that most channels are not expected to dewet, and yet they gate and close. FLYC1 in the up-conformation appears more like a non-conductive state.

Minor points and suggestions

L24: 'FLYC1 is a larger protein and ... '

L29: consider 'that regulate ion preference and conduction'

L42: '...has been studied for decades in plants is the rapid...'

L58: '...that includes five orthologs in E. coli itself (ref Edwards et al., Channels 2012), the eukaryotic plant homologs...'

L79: 'extended by molecular dynamics (MD)...'

L95: please briefly explain the term 'symmetry expanded'

L157-169: in that paragraph describing simulations please mention that tension was NOT applied to the bilayer.

L171: please mention the concentration of the main current-carrying ion under 'physiological recording conditions'

L194-196: the drop of single-channel conductance from 270 to 22 pS in 150 mM KCl implies that there is something more than just electrostatics occurring. If this was electrostatics only, the ionic preference should reverse with a much smaller change in the overall unitary conductance.

L206: it would be interesting to know how the ions behave under the opposite voltage polarity.

L223 and Fig. 4b: the colors of TM5 and TM6 helices in two different states are not easily discernable, one needs to magnify the figure to see the detail. Please increase the contrast.

L254: it would be good to rephrase the statement about inactivation. 'WT FLYC1 does not exhibit a tension-dependent inactivation characteristic of EcMscS (please present a WT FLYC1 trace illustrating non-inactivating behavior in the supplement). The R334E mutation does not alter this behavior.'

L257: I wonder if a disulfide cross-link between these positions is feasible.

L286: please add 'at -425 mV'. Otherwise this statement about Cl⁻ permeability is very incomplete.

L302-305: the fact that the salt bridge is formed in the up-conformation and it stabilizes the non-conductive state may suggest that the up-position is characteristic of a deactivated or resting state.

L308: What are 'pore-proximal' TM helices? Are these pore-forming or pore-lining helices? Or these are TM5's that surround the pore? 'Pore-proximal' is not a good descriptor.

Throughout the previous (Procko et al., *elife* 2021) and this work, FLYC1 has been stimulated with tension generated by a pressure gradient across the patch membrane, which is a typical setting for stretch-activation. Does FLYC1 experience the same activating perturbation *in vivo*? The elongated cells in the indentation zone of a trigger hair have a very special geometry. The channels reside in radially-oriented and peripheral walls (Procko et al., Fig. 3c,d), which are prone to stretching on one side and compression on the other when the trigger hair bends. It appears that membrane stretch produced by cell deformation is indeed the natural stimulus for FLYC1. Some statement about the stimulus in the native setting might be added to the Discussion.

Reviewer #1 (Remarks to the Author):

This study by Jojoa-Cruz, Saotome et al. presents the first structures and associated molecular dynamics and patch-clamp recordings of the mechanosensitive ion channel Flycatcher1 from the Venus flytrap. FLYC1 is related to bacterial MscS and plant MSLs, but has different functional properties (including a higher Cl⁻ selectivity, lower single channel conductance, and lower mechanical threshold for activation) that might be accounted for by unique structural elements. The structures show a generally conserved architecture with new features including additional transmembrane helices and a unique cytoplasmic TM4-5 linker that adopts “up” or “down” conformations in different protomers within the heptameric channel. Molecular dynamics suggests the structure is weakly conductive, perhaps corresponding to a transition or sub-conductance state in the channel gating scheme. Residues in the cytoplasmic cage side portals are shown to be important for conductance and the TM4-5 up conformation is shown to stabilize a closed state. Overall, this is a very interesting and well executed study of this newly identified channel that sets the stage for future studies of gating and determinants of specific properties. I have only minor comments and suggestions for the authors consideration.

1. Is there a reason the authors chose to use an “all-up” structure for molecular dynamics rather than the “6 up 1 down” composite structure? As far as I understand, the 6 up 1 down composite was judged to best represent observed particle conformations in the C1 reconstruction. Perhaps MD on the composite could be performed and compared or connections between the MD and structural data qualified to reflect the difference (for example in the discussion “We did not observe any complete Na⁺ permeation events in our simulations at this timescale, suggesting that our structure of FLYC1 does not represent a fully open conformation”).

We performed additional simulations based on the “6 up 1 down” composite structure, both in a position-restrained (applied to backbone atoms) and a fully unrestrained system. At the start of the simulations, the minimum radius of the pore pathway of the “all-up” structure (2.8 Å) was lower than that of the “6 up 1 down” structure (3.1 Å). Furthermore, simulations starting from the “6 up 1 down” state result in a more consistent pore constriction geometry and are on average very close in minimum radius at the constriction to the initial dimension. However, there is overlap between the pore constriction radius distributions between the various simulation (see new Supplementary Figure 7), especially when the fully unrestrained simulation is considered. Furthermore, no Na⁺ permeation (i.e. channel end to end) events are seen in any of the simulations, whatever the starting structure or the presence/absence of restraints. This suggests that with respect to ion permeation and selectivity estimates, the simulations are robust to changes in initial structure and simulation conditions. We have added description of this results to the main text (lines 267-280).

2. Is there experimental evidence for subconductance states that could correspond to that observed by MD? It is not obvious (to me) from the histogram in 3C.

We have occasionally observed sub-conductance states in WT FLYC1 single channel activity. To conclusively describe a sub-conductance state, several stable long single channel recordings are required, which is often challenging with stretch activated currents because the continuous application of high pressure can lead to patch disruption over time. Without strong data we hesitate to correlate the state in MD simulations to a physiological sub conductance state.

Interestingly, Liz Haswell's group has observed subconductance in MSL10 single channel recordings at more negative membrane potentials, but the frequency of the state was not characterized (Maksaev and Haswell, 2012). To account for this uncertainty, we have labeled our structure as 'near-closed' conformation but included the possibility of it representing a sub-conductance state (line 347). It is possible that *in vivo* and under membrane tension/physiological conditions, the structure is slightly different than isolated protein and exhibits a 'fully-closed' state.

3. It is striking that the pore diameters of FLYC1 and closed EcMscS (and AtMSL1?) are similar and both have phenylalanine ring constrictions, but only EcMscS is found computationally to have a hydrophobic barrier. Could the authors elaborate on chemical differences in the pore that could account for this? Perhaps labeling the residues or positions represented by boxes in S5 would make this easier to interpret?

In closed EcMscS structure, residues L105 and L109 form a 'vapor lock' that serve as a hydrophobic barrier to ion permeation. In FLYC1, V568 and F572 (V319 and F323 in AtMSL1, respectively) are located at similar positions in the pore helix. However, widening of the pore at the level of these two residues in FLYC1 and MSL1, accompanied by a reduction in local hydrophobicity, leads to a hydrophobic barrier not being predicted in these channels.

We have labeled the important residues in Supplementary figure 5a-c and added this interpretation to the manuscript (lines 151-157). Furthermore, we have added Supplementary Fig. 4c, which presents the calculated electrostatic potential for FLYC1, EcMscS and AtMSL1 and labeled the aforementioned residues to aid in the visualization.

4. Do the charge swap mutations K606 and K624 shown to reduce conductance also alter channel selectivity? If not, can the authors speculate on how Cl⁻ selectivity is achieved?

The double mutant K606, K634 does exhibit altered selectivity ($P_{Cl}/P_{Na} = 3.2$) relative to the WT ($P_{Cl}/P_{Na} = 9.8$) (Procko et al., eLife 2021). We have included an I-V curve for the double mutant in Supplementary Figure 8, and value has been reported in the results section (line 223).

5. Is there any evidence the additional TM helices in FLYC1 contribute to functional differences between FLYC1 and EcMscS/AtMCL1?

We do not have any experimental evidence to suggest a contribution of TM1-TM3 to channel function and at this point we can only speculate. For example, additional TMs can alter the change in cross-sectional area between open and closed states, thus potentially altering the free energy difference between the two states and the open probability. Nonetheless, given the lack of experimental evidence, we have decided against inclusion of this speculation in the manuscript.

6. Page 7: Is the question mark in the section title "A phenylalanine ring forms the pore gate of FLYC1?" intentional?

It was intentional. Since we proposed that the FLYC1 structure solved here corresponds to a near-closed conformation, we left the claim of the F572 ring as a question to take into account the possibility of the pore gate corresponding to a different set of residues in the fully-closed

conformation. However, we have changed the section title to “A phenylalanine ring potentially forms the pore gate of FLYC1” to avoid confusion (line 137).

7. Page 7: “van der Waal” should be van der Waals

Thank you for catching this mistake. It has been corrected in the manuscript (line 140).

8. Figure 3b is missing a time scale bar.

Thank you for pointing this out. Scale bars for figure 3b&c have been added.

Reviewer #2 (Remarks to the Author):

Jojoa-Cruz et al. present an important biophysical study that may reveal the fascinating molecular mechano-sensation mechanism of Venus flytraps, by combining structural biology, computational biology, and electrophysiology approaches. Although the overall structure of the MS ion channel Flycatcher1 is similar to previously solved homologous proteins such as MscS and AtMSL1, a new structural feature of the TM4-TM5 linker was observed, which may be related to a distinct gating mechanism. The MD simulations and electrophysiology experiments significantly strengthened the manuscript, providing atomistic, dynamic, and quantitative information. The manuscript is well written, clear and concise.

I have several questions/suggestions for the authors to address:

1. Can the TM4-TM5 linker reach the side portal in the 'down' conformation?

The TM4-TM5 linker is nearby the side portal in the down conformation but, in our model, it does not appear to contact it. It is possible that the position of the linker plays a role in the local concentration of ions, it is not feasible to determine it at this point given that there is unmodelled density surrounding the linker (see Fig. 1a and Supplementary Fig. 9c). It is also possible that part of this unmodelled density contacts the side portal.

2. Why does the 'down' conformation (or disruption of an up conformation-specific interaction) stabilize the open state? The authors ought to give some discussions on the likely underlying reason at least.

We have added the following sentence in discussion section (line 363): “We speculate the R334E mutant channel remains in an open conformation for a longer duration because the mutation increases the energetic barrier for the transition from open to the closed state. This could be due to an additional interaction in the open state by the introduced E334 residue or the R334-D598 interaction favoring the closed state.”

3. The mutation D598K led to a non-functional channel, why? Since the authors state that breaking the salt bridge between D598 and R334 should stabilize the open state / slow down the deactivation, as evidenced by the mutant R334E, one may expect that D598K would have a similar effect. The "unexpected" non-functional mutation D598K should be clarified.

As discussed in the comment above, we have included the possibility of R334E mutation stabilizing the open conformation by an unknown interaction in the open state. In such scenario, it would be expected that the D598 mutation does not exhibit the same phenotype. Alternatively, the phenotype may be due to improper trafficking or folding of the D598K mutant. We have previously reported similar issues with AtMSL10 overexpression in mammalian cells (Procko et al., eLife 2021), where no MA currents could be recorded despite these channels exhibiting mechanosensitive behavior in other systems (oocytes) (Maksaev and Haswell, 2012). We have added this clarification to the manuscript (line 363-371).

4. The Y axes in Fig 2b and 2c are in the reverse directions, which makes it a bit difficult to understand the location in the pore, especially when Fig 2a-c are put together. I suggest the authors use the same y-axis direction and range for Fig 2b and 2c.

We thank the reviewer for catching this detail. The point of reference for one of the axes was inverted (positive values should be negative and vice versa). We have corrected this to have both figures with the same direction. Regarding the range, we believe having the expanded range in Fig. 2b does add value to understand the pore profile, but we have added a box to represent the area that is comparable to Fig. 2c.

5. I would like to see a video showing a complete Cl^- permeation event in the MD simulations, which would be appealing to the audience too.

A typical end-to-end chloride ion crossing event (in the efflux direction) is now presented in Supplementary Movie 1.

6. How dynamic is the TM4-TM5 linker in the MD simulations?

The dynamics of the TM4-TM5 linker were initially not explored in our backbone position-restrained simulations. We have now performed an additional unrestrained simulation and attached a root mean square fluctuation (RMSF) plot for the $\text{C}\alpha$ atoms in Supplementary Figure 10a. The TM4-TM5 linker (intracellular side), and to a lesser extent the TM3-TM4 linker (extracellular side) are two regions of higher conformational flexibility. We do however note that such flexibility of the TM4-TM5 linker should be viewed with caution given there are ~150 unmodelled residues within this loop (line 275-280).

Further, the dynamics of the linker as a function of time was further analyzed in Supplementary Figure 10c-d. The system equilibrates in ~20 ns into the simulation with some subunits alternating between the up and down states.

7. How is the dynamics of the TM4-TM5 linker affected by the mutation R334E/D598K? These MD simulations/analyses may be informative for understanding the role of the TM4-TM5 linker. We did not run simulations with the R334E or D598K mutations, so we cannot conclusively comment on how these mutations would alter the dynamics in the simulations. We hypothesize the differences may not be as large as expected, as electrophysiology results suggest the differences between mutant and WT are observed after mechanical stimulus, which we do not incorporate in our simulations.

Reviewer #3 (Remarks to the Author):

The manuscript ‘Structural Insights into the Venus flytrap Mechanosensitive Ion Channel Flycatcher1’ by Jojoa-Cruz and coworkers provides the first data on the structure and mechanism of this intricate version of an MSL-like channel that helps carnivorous plants detect prey. The cryo-EM structural information on the mostly resolved central core domains is extended by MD simulations, which estimate the dynamics and conductance of the pore in one of the conformations (up-state). The structure and simulations give important functional predictions regarding the ionic selectivity determinants and open-state stabilizing interactions. These predictions were supported by mutagenesis and patch-clamp experiments.

In the cryo-EM section, the authors judiciously applied the seven-fold symmetry constraints only to certain 2D classes at selected stages of analysis and treated others as C1 objects which allowed them to discern asymmetric conformations with ‘up’ and ‘down’ positions of the TM4-TM5 cytoplasmic linker in different protomers. This advanced approach paid off in identifying the minor ‘down’ component, which was functionally designated as a step toward opening. The technical aspects of this work raise no questions apart from my request to better explicate the concept of symmetry expansion. This methodology is distinct from most of the previous studies of MscS-like channels where C7 symmetry was applied by default.

We agree that treatment of MscS-like channels as non-symmetric channels is distinct from most of the previous studies, however, the use of symmetry expansion is not a new method in cryo-EM processing generated by our study. We have provided a short explanation in the methods and referenced Zhou, et al. (Genes & development, 2015), where details of the method are reported (line 461). Furthermore, strict symmetry enforcement (e.g. C7) is typically applied to improve the resolution of cryoEM maps, but does not necessarily represent the true biological state of a protein. Symmetry expansion allows more flexibility and as the reviewer notes, enabled us to identify the relatively minor “down” state.

My major reservation concerns the assignment of FLYC1 in the up-state as a subconductive conformation. The authors state that, based on the heuristic assessment, the gate of the channel has no tendency to dewet. It would be important to show a water density plot along the z axis as a function of time at zero voltage.

We have indeed performed simulations initially in the absence of electric field and noted that the pore has no tendency to de-wet (line 170). Our decision to subsequently apply an electric field was driven by the observation of more ion crossing events. The location of the water in the z-axis, in the absence of electric field, is now presented in Supplementary Figure 6 as a density plot and a time series.

It might be also good to justify the choice of TIP4 water in the Methods.

A number of simulation studies have compared water models used in simulation studies of ions and ion channels, some examples are highlighted below. From these it is evident that the widely used TIP3P model overestimates both water and ion diffusion rates, The TIP4P model is a reasonable compromise in terms of accuracy vs. computational cost – future studies could

explore polarizable models (see e.g., Klesse *et al.*, referenced below, also recent paper from Benoit Roux's Lab (Villa *et al.*, *J Phys Chem A* 2018, 6147–6155)).

Simple comparisons

H. Zhang, C. Yin, Y. Jiang, and D. van der Spoel. “Force Field Benchmark of Amino Acids: I. Hydration and Diffusion in Different Water Models”. In: *Journal of Chemical Information and Modeling* 58.5 (2018), 1037–1052.

Ion channel comparison

Klesse, G.; Rao, S.; Tucker, S. J.; Sansom, M. S. P., Induced polarization in molecular dynamics simulations of the 5-HT₃ receptor channel. *J. Amer. Chem. Soc.* **2020**, *142*, 9415–9427.

Nanopore comparison

Calvelo, M.; Lynch, C. I.; Granja, J. R.; Sansom, M. S. P.; Garcia-Fandiño, R., Effect of water models on transmembrane self-assembled cyclic peptide nanotubes. *ACS Nano* **2021**, *15*, 7053–7064.

Ngo, V.; Li, H.; MacKerell, A. D.; Allen, T. W.; Roux, B.; Noskov, S., Polarization effects in water-mediated selective cation transport across a narrow transmembrane channel. *J. Chem. Theor. Comput.* **2021**, *17*, 1726–1741.

Fig 2a depicts some sort of water-accessible cross-section along the pore and only selected sidechains are shown (with no volume). Because TM6 helices in FLYC1 are packed less tightly than TM3s in EcMscS, it is expected that more water will be in the interhelical spaces and more polar atoms exposed to the pore water. For this reason, the authors should consider presenting the inner surface of a vertical pore cross-section colored by atom polarity.

Supplementary Fig. 4c has been added to show that looser packaging of TM6 in FLYC1 causes a widening of the pore towards the extracellular side (Fig. 2b) and a larger surface of polar residues to be exposed and lining the pore in the TMD relative to EcMscS.

The energy of a fully solvated Cl⁻ ion was not assessed along the path and only the number of ions passing the pore per ns was counted. The voltage of -425 mV is a concern since non-linear effects might be present and dielectric water attraction into the region of high electric field (pore constriction) is expected. What is the Cl⁻ permeation rate at 100 and 200 mV? In their 2019 PNAS paper Sansom and colleagues show that most channels are not expected to dewet, and yet they gate and close. FLYC1 in the up-conformation appears more like a non-conductive state. We have now performed additional simulations (with the “6 up 1 down” model) at potential differences of -450 mV, -225 mV and -112.5 mV, to assess whether the wetting behavior of the pore is affected by the strength of the electric field. The calculated conductance (based on single 100 ns simulations) is in the range of 130–250 pS. In particular, these simulations do not provide any evidence of electric-field induced wetting, nor any marked non-linear correlation between the applied electric field and the ion flux. We previously estimated the conductance to be ~50 pS based on simulations on the “all up” structure. The estimation based on the “6 up 1 down” structure (average across simulations of ~190 pS) is in fact closer to the experimentally estimated

range (164–276 pS). A summary of the calculation and the count of ion crossings is now presented in Supplementary Table 2.

Minor points and suggestions

L24: ‘FLYC1 is a larger protein and ...’
The suggested change has been made (line 25).

L29: consider ‘that regulate ion preference and conduction’
The suggested change has been made (line 30).

L42: ‘...has been studied for decades in plants is the rapid...’
The suggested change has been made (line 39).

L58: ‘...that includes five orthologs in E. coli itself (ref Edwards et al., Channels 2012), the eukaryotic plant homologs...’
The suggested change has been made and reference included (line 55).

L79: ‘extended by molecular dynamics (MD)...’
The suggested change has been made (line 78).

L95: please briefly explain the term ‘symmetry expanded’
Please see first comment to reviewer #3.

L157-169: in that paragraph describing simulations please mention that tension was NOT applied to the bilayer.
This detail has been added in line 166.

L171: please mention the concentration of the main current-carrying ion under ‘physiological recording conditions’
We have described the ionic composition of the physiological recording condition in the text.
Line 189: “in physiological recording solution that contains 130 mM NaCl, 5 mM KCl, 1 mM CaCl₂, and 1 mM MgCl₂...”

L194-196: the drop of single-channel conductance from 270 to 22 pS in 150 mM KCl implies that there is something more than just electrostatics occurring. If this was electrostatics only, the ionic preference should reverse with a much smaller change in the overall unitary conductance. We were surprised by the large reduction in conductance as well. Our now added selectivity data indicates that the double mutant has a threefold lower chloride to sodium permeability ratio compared to WT channels (line 222). Therefore, these mutations could be simply decreasing the drive for the ions. It is possible that in WT FLYC1 channels lysine residues in the side portal increase local chloride concentration, whereas in the double mutant the higher negative charge could repel chloride ions resulting in a smaller conductance.

L206: it would be interesting to know how the ions behave under the opposite voltage polarity. When positive potentials were applied, we see chloride ions entering the channel via the central pore, but the majority of these ions remain in the cytoplasmic vestibule throughout repeats of our 100-ns simulations, interacting with charged & polar residues facing the vestibule interior. As these are not complete passage events through the entirety of the channel, the estimated conductance based on ion crossing events is c.a. 1-fold lower than when negative potentials were applied. It is conceivable that the vestibule can become saturated with ions and achieve an equilibrium, therefore longer extended simulations in the μs range in our future work would delineate the behavior of ions (and that of sodium ions too) under the influence of electric fields in both directions.

L223 and Fig. 4b: the colors of TM5 and TM6 helices in two different states are not easily discernable, one needs to magnify the figure to see the detail. Please increase the contrast. Thank you for the suggestion. We have increased the contrast for the two TMs in Fig. 4b.

L254: it would be good to rephrase the statement about inactivation. 'WT FLYC1 does not exhibit a tension-dependent inactivation characteristic of EcMscS (please present a WT FLYC1 trace illustrating non-inactivating behavior in the supplement). The R334E mutation does not alter this behavior.'

Thank you for the suggestion, we have now rephrased the inactivation sentence (now line 300). Figure 4d that displays representative traces for FLYC1 WT and R334E mutant already has an inset with an expanded view of the two traces. This example highlights that the two channels activate and inactivate similarly.

L257: I wonder if a disulfide cross-link between these positions is feasible.

We think this is a great suggestion, one that has occurred to us too. Although this is an intriguing experiment, it is beyond the scope of our current study. In addition to mutagenesis, these experiments will involve several fitting controls associated with MTS treatment (for example, single cysteine mutations only and test for reversibility with reducing agents), not to mention that technically these experiments can be challenging for MA ion channels. Nonetheless, in the future, us and other groups in the field can use our FLYC1 structure to test and build upon this idea to determine whether acutely stabilizing the "down" conformation with disulfide cross-links can affect channel function in real time.

L286: please add 'at -425 mV '. Otherwise this statement about Cl^- permeability is very incomplete.

"at -425 mV " has been added (line 338).

L302-305: the fact that the salt bridge is formed in the up-conformation and it stabilizes the non-conductive state may suggest that the up-position is characteristic of a deactivated or resting state.

This point is partially addressed in the second part of comment 1 from reviewer #2. Additionally, we have added the following sentence to the discussion (line 374): “Likewise, the stability of the closed state in the WT where this salt bridge is, presumably, formed suggest that the up conformation may be favored in the resting state.”

L308: What are ‘pore-proximal’ TM helices? Are these pore-forming or pore-lining helices? Or these are TM5’s that surround the pore? ‘Pore-proximal’ is not a good descriptor.

We replaced “pore-proximal” with “TM3-TM6 helices” (line 374).

Throughout the previous (Procko et al., eLife 2021) and this work, FLYC1 has been stimulated with tension generated by a pressure gradient across the patch membrane, which is a typical setting for stretch-activation. Does FLYC1 experience the same activating perturbation in vivo? The elongated cells in the indentation zone of a trigger hair have a very special geometry. The channels reside in radially-oriented and peripheral walls (Procko et al., Fig. 3c,d), which are prone to stretching on one side and compression on the other when the trigger hair bends. It appears that membrane stretch produced by cell deformation is indeed the natural stimulus for FLYC1. Some statement about the stimulus in the native setting might be added to the Discussion.

The reviewer highlights a very interesting point but we hesitate to make any comparison between the pressure applied in our cellular assay to the tension experienced in the sensory cells. For one, the architecture of both cell types (HEK cells vs. Venus flytrap sensory cells) are very distinct. We think differences in membrane composition and cellular ultrastructure could have an effect on how local tension or force is transduced. Therefore, unless there is evidence that the activation tension of the channel in our cellular assay is similar to the activation tension caused by the bending of the trigger hair, one cannot make those claims.

REVIEWERS' COMMENTS

Reviewer #1 (Remarks to the Author):

The authors have fully addressed all points I raised in review and I have no further comments. I look forward to seeing the paper published.

Reviewer #2 (Remarks to the Author):

The authors have satisfactorily addressed my questions and made necessary changes to the manuscript. I am happy to recommend its publication.

Reviewer #3 (Remarks to the Author):

I checked the revised manuscript 'Structural Insights into the Venus flytrap Mechanosensitive Ion Channel Flycatcher1' and I am satisfied with the revisions. Extremely thorough work.

My only request is to provide the water density units in Supplemental Figure 6b. I am still puzzled by the fact that the water density drops in the narrow region near F572 by about two orders of magnitude compared to the bulk, and yet the pore is capable of conducting at ~200 ps. Both the degree of hydration and conductance somehow stay unaffected in the range of voltages between -112 and - 450 mV.

REVIEWERS' COMMENTS

Reviewer #1 (Remarks to the Author):

The authors have fully addressed all points I raised in review and I have no further comments. I look forward to seeing the paper published.

Reviewer #2 (Remarks to the Author):

The authors have satisfactorily addressed my questions and made necessary changes to the manuscript. I am happy to recommend its publication.

Reviewer #3 (Remarks to the Author):

I checked the revised manuscript 'Structural Insights into the Venus flytrap Mechanosensitive Ion Channel Flycatcher1' and I am satisfied with the revisions. Extremely thorough work.

My only request is to provide the water density units in Supplemental Figure 6b. I am still puzzled by the fact that the water density drops in the narrow region near F572 by about two orders of magnitude compared to the bulk, and yet the pore is capable of conducting at ~200 ps. Both the degree of hydration and conductance somehow stay unaffected in the range of voltages between -112 and -450 mV.

We agree that the previous version of Supp. Fig. 6 can be confusing. The profile shown in the previous Supp. Fig. 6b was a probability density function for water occurrence and as such is effectively a (normalised) convolution of water number density and pore radius. In the revised figure we show instead the more conventional water number density and pore radius profiles derived from the simulation using the analysis described in <https://www.channotation.org/>. This demonstrates the water density in the F572 ring region is not lowered and thus that this region is not de-wetted during the simulations. We think this is now clearer for the reader and apologise for the earlier ambiguity in the labelling of the axis.